# Long-Term Effect of Low-Frequency Electromagnetic Irradiation in Water and Isotonic Aqueous Solutions as Studied by Photoluminescence from Polymer Membrane

**DOI:** 10.3390/polym13091443

**Published:** 2021-04-29

**Authors:** Nikolai F. Bunkin, Polina N. Bolotskova, Elena V. Bondarchuk, Valery G. Gryaznov, Sergey V. Gudkov, Valeriy A. Kozlov, Maria A. Okuneva, Oleg V. Ovchinnikov, Oleg P. Smoliy, Igor F. Turkanov

**Affiliations:** 1Department of Fundamental Sciences, Bauman Moscow State Technical University, 2-nd Baumanskaya Str. 5, 105005 Moscow, Russia; bolotskova@inbox.ru (P.N.B.); v.kozlov@hotmail.com (V.A.K.); neonlight0097@gmail.com (M.A.O.); 2Prokhorov General Physics Institute of the Russian Academy of Sciences, Vavilova St. 38, 119991 Moscow, Russia; s_makariy@rambler.ru; 3“Concern GRANIT”, Gogolevsky Blvd., 31, Bldg. 2, 119019 Moscow, Russia; info@npo-qt.ru (E.V.B.); gryaznov.v@granit-concern.ru (V.G.G.); office@granit-concern.ru (O.V.O.); smoliy.o@granit-concern.ru (O.P.S.); turkanov.i@granit-concern.ru (I.F.T.); 4Department of Biophysics, N.I. Lobachevsky State University of Nizhny Novgorod, 603950 Nizhny Novgorod, Russia; 5All-Russian Research Institute of Phytopatology, Institute St. 5, Bol’shie Vyazemy, 143050 Moscow, Russia

**Keywords:** swelling of polymers, photoluminescence spectroscopy, low-frequency electromagnetic treatment, isotonic solution, long-term effects of electromagnetic treatment, cell membrane, bubstons, bubston clusters

## Abstract

The swelling of a polymer membrane Nafion^TM^ in deionized water and isotonic NaCl and Ringer’s solutions was studied by photoluminescent spectroscopy. According to our previous studies, the surface of this membrane could be considered as a model for a cellular surface. Liquid samples, in which the membrane was soaked, were subjected to preliminary electromagnetic treatment, which consisted of irradiating these samples with electric rectangular pulses of 1 µs duration using platinum electrodes immersed in the liquid. We used a series of pulses with a repetition rate of 11–125 Hz; the pulse amplitudes were equal to 100 and 500 mV. It turned out that at certain pulse repetition rates and their amplitudes, the characteristic swelling time of the polymer membrane significantly differs from the swelling time in untreated (reference) samples. At the same time, there is no effect for certain frequencies/pulse amplitudes. The time interval between electromagnetic treatment and measurements was about 20 min. Thus, in our experiments the effects associated with the long-term relaxation of liquids on the electromagnetic processing are manifested. The effect of long-term relaxation could be associated with a slight change in the geometric characteristics of bubston clusters during electromagnetic treatment.

## 1. Introduction

The study of interactions of nonionizing electromagnetic waves with whole organisms, and specifically with well-characterized cellular model systems in vitro, has received increased recognition. A growing number of experimental findings has been reported, and hypothetic mechanisms which might be involved in mediating these effects have emerged (for review, see [1,2,3,4,5,6,7,8]).

At the same time, studies of the interaction of the low-frequency range with cellular structures are of considerable interest. At the same time, impulse low-frequency action on living organisms is of particular interest. For example, in a recent work [9], it was shown that a low frequency (7 Hz) pulsed electromagnetic field induces cell death in native proliferating cells for a sufficiently wide sampling of patients. Concluding this section, we note that low-frequency electromagnetic exposure has a therapeutic effect, see Table 2 in a recent review [8].

As follows from the above review, the effects caused by the electromagnetic radiation of biological systems manifest themselves at the cellular level; the majority of experiments were carried out for cell suspensions in isotonic solutions. In this regard, it should be noted that some polymer membranes and cell membranes exhibit similar properties. Specifically, in our recent work [10], it was shown that when the Nafion^TM^ polymer membrane swells in water, polymer fibers “unwind” into the water bulk. It is important that these fibers do not completely tear off the membrane interface, i.e., a “brush”-like structure close to the interface is generated. Furthermore, when Nafion swells in water, a network of through channels in the membrane volume is formed (see the review [11] for more detail). The presence of such channels has been applied in a number of practical applications. For example, the processes of liquid transfer through the capillary network of such channels are widely studied with taking into account their fractal properties and the surface roughness of the membrane, see, for example, [12,13].

An important parameter for Nafion-based fuel cells is the swelling rate of the membrane in water. In this regard, it is necessary to mention the work [14], where the complex environments experienced by water molecules in the hydrophilic channels of Nafion membranes were studied by ultrafast infrared pump-probe spectroscopy. Time-dependent anisotropy measurements showed that the orientational motions of water molecules in the channels of Nafion membranes are significantly slower than in bulk water and that lower hydration levels result in slower orientational relaxation. This behavior was analyzed using a model based on restricted orientational diffusion within a hydrogen bond configuration followed by total reorientation through jump diffusion.

Bearing in mind that the membrane is “decorated” with unwound polymer fibers, we arrive at an analogy with the cell membrane. Indeed, the structure of the channels in the bulk of polymer is similar (conditionally) to the lipid bilayer of the cell membrane, while the external structure of the polymer fibers is similar to the glycocalyx (extracellular matrix, see [15]). This analogy was developed on a qualitative level in our recent work [16]; Within the approach based on this analogy a number of features revealing at swelling of polymer membranes in water and aqueous salt solutions have been explained. Therefore, there is a natural interest in investigating the specific interaction of Nafion with deionized water and various isotonic solutions, which were subjected to the low-frequency electromagnetic irradiation.

As shown in [16], a number of effects associated with the swelling of the Nafion membrane in water and aqueous solutions of salts (in [16], aqueous solutions of NaCl were studied in a wide range of concentrations) are due to the presence of a nanobubble phase in water and aqueous solutions of electrolytes (for more details see Section 4 below). Furthermore, it was found in [16] that the swelling kinetics of the Nafion membrane has a long-term relaxation effect. Namely, effective shaking of a liquid sample (deionized water) leads to a change in the kinetics of membrane swelling, and this effect has a significant relaxation time (about a day). 

The present work is devoted to the study of the effects of long-term relaxation in water and aqueous salt solutions, resulted from the processing of liquid samples with electromagnetic pulses at different pulse repetition rates and amplitudes.

## 2. Materials and Methods

### 2.1. Materials

Polymer membrane Nafion (C_7_HF_13_O_5_S × C_2_F_4_) is a sulfonated tetrafluoroethylene based fluoropolymer-copolymer. The polymer matrix consists of a tetrafluoroethylene backbone, where perfluorovinyl ether groups are terminated with sulfonate groupsHSO_3_.We investigated NafionN117 plates (Sigma Aldrich, St. Louis, MO, USA) with a thickness of *L*_0_ = 175 μm. The Nafion plates were soaked in Milli-Q water with a resistivity of 4 MΩ⋅cm (measurement were made 1 h after the preparation) in isotonic NaCl (0.9%; Mosfarm, Russia) and Ringer’s (Medpolymer, Russia) solutions. In our particular case, the Ringer’s solution was composed of NaCl (8.6 g/L), KCl (0.3 g/L), and CaCl_2_ × 6H_2_O (0.25 g/L), dissolved in water.

### 2.2. Instrumentation

#### 2.2.1. Photoluminescence Study

In this subsection we briefly describe an experimental protocol; for more detail see [10]. The experimental technique is based on the excitation of photoluminescence from the Nafion surface by pumping in the UV range. To excite luminescence, it is necessary to irradiate a substance within one of the absorption bands. Some remarks, germane to the issue, are these: It is known [17] that one of the maxima of the absorptivity of both dry and swollen in water Nafion occurs at a wavelength of λ = 270 nm. It is also known (see, for example, Ref. [18]) that water does not absorb in this spectral range. In further support, we note that when optical radiation interacts with matter, and a complex molecular system is irradiated inside the absorption band, quantum transitions are stimulated in a longer-wave range, i.e., photoluminescence should arise. 

The present work is devoted to the study of the interaction between the Nafion membrane and liquid samples, processed with electric pulses at low frequency of repetition, by photoluminescent techniques, which are widely exploited in polymer studies, see, for example, [19,20,21,22,23]. In our photoluminescence experiments we used radiation at a wavelength of λ = 369 nm corresponding to the long-wavelength region of the absorption band. In parallel experiments (see Ref. [10]), it was found that the terminal sulfonic groups of HSO_3_ serve as the centers of Nafion luminescence under UV irradiation.

Figure 1 illustrates the dependence of the luminescence signal *P* from the Nafion solution in isopropanol at a wavelength of λ = 460 nm (the spectral maximum of the luminescence); The zero abscissa corresponds to pure isopropanol. The concentration of the solution in this case is not known to us; we consider (conditionally) that the concentration is 100 a.u. immediately after removal of the Nafion plate from the solution, and then the solution was diluted proportionally with isopropyl alcohol.

The rectilinear segment in Figure 1 can be approximated by the following formula: *P* = −237 + 16 *n_Naf_*(1)
where *n_Naf_* is the volume number density of the luminescence centers, i.e., terminal sulfonic groups. Since these groups are attached to polymeric chains, *n_Naf_* can be associated with the volume number density of Nafion particles. The dependence obtained can be represented as:*P = A + kI_pump_**σ_lum_n_Naf_V,*(2)
where *I_pump_* is the pump intensity, *A* = −270 relative units corresponding to the spectral density of the mini-spectrometer noise and stray-light illumination, *k* is the transfer coefficient of the setup, *V* is the luminescence volume, and *σ_lum_* is the luminescence cross section (it is obvious that the spectral maximum of *σ_lum_* corresponds to λ = 460 nm). It follows from Figure 1 that *σ_lum_* = const.

To carry out experiments on photoluminescence spectroscopy, a photoluminescence spectroscopy setup in grazing incidence geometry was developed and assembled (for more detail see Ref. [10]). The schematic of the experimental setup is shown in Figure 2.

The probing radiation of continuous wave laser diode (1) (optical pumping) at a wavelength λ= 369 nm was introduced into the multimode quartz optical fiber (2) with a diameter Φ = 100 μm and a numerical aperture *NA* = *n* × sinα = 0.3, where *n* = 1 is the refractive index of air, α is the angle of beam divergence at the exit end of the fiber in air. The fiber was fixed in a hole located in the center of the bottom of a cylindrical cell (3) made of stainless steel; the direction of pump beam set the optical axis of the experimental setup. The cell was thermostabilized at room temperature (*T* = 23 °C) accurate to ± 0.1 °C and filled with test liquid sample. We studied the swelling of a square Nafion plate (4) with a side *h* = 4 mm and a thickness *d* = 175 μm. The plate was fixed parallel to the optical axis, i.e., the experiments were carried out at grazing incidence. The vertical edges of the Nafion plate were rigidly fixed with two vertical parallel clamps in parallel to the optical axis, i.e., the experiments were carried out at grazing incidence; no additional substrate for fixation of the plate was used. The size of the clamps was much less than the width of the Nafion plate, i.e., for the central region of the plate along the vertical direction (this region was subject to irradiation by the UV pumping), the approximation of the free boundary was realized.

At the beginning of the experiment, a dry (water-free) Nafion plate was fixed in an empty cell; the plate could be displaced horizontally using a stepping motor (8), thereby changing the position of the plate in relation to the optical axis. In doing so, we achieved the maximum of the luminescence signal; the corresponding position of the Nafion plate relative to the optical axis is considered to be optimal. When a liquid sample was poured into the cell, the initially hydrophobic Nafion plate was bent along the optical axis. However, such bending led only to an effective shift of the Nafion-water boundary (this shift amounted to about 1 mm) but did not result in a change in the incident angle of the pump radiation. To restore the optimal position of the plate relative to the optical axis, additional adjustment was carried out using a stepping motor (8). The luminescence radiation was reflected by the internal surface of the cell (Nafion is transparent in the visible range) and collected along the optical axis of the cell. This resulted in a significant gain in luminescence intensity. The luminescence signal was received by a quartz fiber (5), fixed at the center of the cell and transferred to the minispectrometer (6) FSD-8 (Russia). The experimental data were accumulated by a computer (7). In the experiment, the temporal dynamics of the luminescence intensity in the spectral maximum depending on the time of soaking of the Nafion plate in the liquid under study was investigated. The start of soaking time counting corresponds to the moment of pouring the liquid sample into the cell.

#### 2.2.2. Processing of Liquid Samples

To study the effects of electromagnetic treatment of the test liquids, we designed an experimental setup which allowed us to expose liquid samples to electrical pulses of positive/negative polarity with a repetition rate in the range of 5–125 Hz with the possibility of varying the amplitudes and durations of the pulses. In these experiments, we used a G5-63 pulse generator (Russia), and an oscilloscope AKIP 4115/3A (Russia).

The test liquid samples were poured into a 5-mL cell. Next, platinum electrodes connected to a pulse generator were immersed into the cell; the distance between the electrodes was 5 mm, and the area of the electrodes in the liquid was 10 mm^2^. We used pulses with durations of μs and amplitudes of 100 mV and 500 mV (the voltage data was taken after immersing the electrodes in a liquid sample). Liquid samples were exposed to electrical impulses for 20 min (the time of processing). Typical pulses used in our experiment are shown in Figure 3a—100 mV and Figure 3b—500 mV; the pulse repetition rate is indicated in the lower right corner. The electrochemical potential of platinum in accordance with the reaction Pt^2+^ + 2*e*^−^ ⇌ Pt (*s*) is φ_0_ = +1.2 V, see Ref. [24]; it is obvious that the voltage of pulses supplied to the electrodes should not exceed the value of *φ*_0_ just to avoid electrochemical reactions on the electrodes. To make sure that we did not have to deal with electrochemical reactions, we controlled the pH of the test liquids before and after electromagnetic processing. For that purpose, we used pH-meter HANNA HI 98108 PHep+ (USA), which was calibrated with standard acid and alkaline titers, having pH = 4.06 and 9.18 accordingly. It turned out that the electromagnetic processing did not result in the change of pH: before and after processing for deionized water the pH = 5.7 ± 0.1, for NaCl solution pH = 6.2 ± 0.1, for Ringer’s solution pH = 5.5 ± 0.1.

## 3. Experimental Results

The dependences of the luminescence intensity at the spectral maximum vs. the swelling time of the polymer membrane, presented below, correspond to averaging over five successive measurements. Confidence intervals are indicated on all graphs, see below.

First, we measured the luminescence intensity at the spectral maximum (460 nm) as a function of the soaking time for unprocessed (reference) samples of deionized water after settling for an hour in a flask open to the atmosphere, as well as unprocessed (reference) samples of isotonic NaCl solution and Ringer’s solution. Figure 4a shows the time dependence of the photoluminescence intensity at the spectral maximum (460 nm) for deionized water and NaCl solution. Figure 4b exhibits the time dependence of the photoluminescence intensity for Ringer’s solution. We can see that for all the liquids, the experimental points belong to approximately the same exponential curves. The corresponding formulas are given in the insets and highlighted with different colors: blue color for water and red color for NaCl solution. As follows from Figure 4a, the decay times of the exponential function for deionized water and NaCl solution are the same and equal to τ_0_ = 14.3 min, and for Ringer’s solution (Figure 4b we have τ_0_ = 12.3 min. The pre-exponential factors and free constants for the given exponentials also insignificantly differ. Thus, for untreated deionized water and isotonic salt solutions the time dependences of the luminescence intensity are well approximated by very close decaying exponential functions, i.e., the ionic additives would hardly influence on the dynamic of swelling Nafion.

The time interval between the finish of electromagnetic processing of the liquid samples and the start of measuring the photoluminescence intensity amounted to 10 min. This interval included the initial alignment of the setup for the photoluminescence spectroscopy followed by pouring the processed liquid into the cell shown in Figure 2.

In order not to clutter up the article, we present only the most characteristic dependences of the luminescence intensity on the time of soaking of the polymer membrane. In Figure 5 we exhibit the dependence for Ringer’s solution treated with pulses with an amplitude of 100 mV and a repetition rate of 20 Hz. In the same figure we show the dependence for the reference sample. It can be seen that both dependences are well approximated by decaying exponential functions, but for the processed sample the decay time is approximately two times less. The same graph shows the calculations of the Pearson’s rank correlation coefficient for the obtained dependences. The value of rank correlation coefficient (0.9) shows that in both cases we are apparently dealing with a process that has the same physical nature, but the kinetic characteristics of this process differ after processing with electric pulses. Note that the resulting graphs were averaged over five successive dependencies: the graphs indicate the confidence interval for the points obtained by averaging.

In Figure 6 we display the dependence for Ringer’s solution treated with pulses with an amplitude of 100 mV and a repetition rate of 125 Hz; here we also show the dependence for the reference sample. It can be seen that both dependences are also well approximated by decaying exponential functions, but for the processed sample the decay time is approximately twice as long.

In Figure 7 we display the dependence for Ringer’s solution treated with pulses with an amplitude of 100 mV and a repetition rate of 100 Hz. The same graph shows the dependence for the reference sample. It is seen that both dependences are well approximated by decaying exponential functions and approximately coincide.

As follows from these graphs, after processing with electric pulses, the decay time of the exponential functions, used for approximating the experimental dependences, may increase/decrease in comparison with the reference dependences, or may not change at all. The results of the experiments performed are summarized in Figure 8, where we exhibit the dependences of τ/τ_0_, where τ_0_ is the decay time of the reference dependence, τ is the decay time of the corresponding dependence after processing, on the pulse repetition rate for deionized water (Figure 8a), isotonic NaCl solution (Figure 8b), and Ringer’s solution (Figure 8c); black curves correspond to the pulse amplitude of 100 mV, and red curves correspond to 500 mV pulse amplitude. It is seen that the obtained dependences are qualitatively correlated to one another for all investigated liquids.

## 4. Discussion

As follows from the graph in Figure 1 for the luminescence intensity at the spectral maximum (see also Equation (1)), the dynamics of the luminescence intensity *I* is completely controlled by the volume number density of the luminescence centers *n_Naf_* (sulfonic groups in our case, see [10]), the luminescence cross section *σ_lum_* and the luminescence volume *V*. The volume number density of luminescence centers in volume V can be represented as *n_Naf_* = *C/V*, where *C* is the total amount of the polymer particles in the volume *V*. Obviously, this volume is determined by the cross section of the laser beam in the near-surface layer of the Nafion plate (recall that we irradiate the plate in grazing incidence geometry), and the plate height, i.e., the volume *V* is fixed. Since water molecules penetrate the surface layer of the membrane during swelling, the *n_Naf_* value inside this layer should decrease. Then, under the assumption that σ*_lum_* is constant (note that there is no reason to assume that σ*_lum_* changes at swelling), the only time-dependent parameter is the bulk density of luminescence centers *n_Naf_*. Introducing the characteristic decay time τ, which is about the swelling time, we obtain the differential equation:(3)$$\frac{dn_{Naf}}{dt}=-\frac{n_{Naf}}{\tau }$$
i.e.,
(4)$$n_{Naf}={\left(n_{Naf}\right)}_{0}exp\left(-\frac{t}{\tau }\right)$$
where (*n_Naf_*)_0_ is the volume number density of luminescence centers at *t* = 0 (i.e., in dry Nafion). As follows from the above experimental plots, processing with electrical pulses leads to a change in the swelling time τ.

As was indicated above, the Pearson’s rank correlation coefficient between the reference and processed liquid samples is close to unity. In our opinion, this indirectly suggests that the luminescence cross section σ*_lum_* does not change as a result of processing, that is, the dynamics of membrane swelling for treated and untreated samples is described by exponential Equation (4). It is very important that for the reference samples the exponential functions describing the swelling dynamics are practically the same, see Figure 4. At the same time, as is seen from the graphs in Figure 8, the characteristic swelling time τ changes after the treatment with electric pulses.

As is seen from the graphs, the ratio τ/τ_0_ for different samples depends significantly on the repetition rate and amplitude of the pulses. For example, for isotonic NaCl solution (panel (b)), treatment at frequencies of 11 and 20 Hz for the pulse amplitude of 100 mV leads to a significant increase in the swelling rate of the polymer membrane, while for the pulse amplitude of 500 mV, the swelling rate very slightly changes compared to the reference sample. Note also that the most significant increase in the swelling rate is also observed for an isotonic NaCl solution at a repetition rate of 125 Hz for both pulse amplitudes. Interestingly, the pulse treatment of deionized water for both pulse amplitudes leads primarily to a slowdown, rather than an increase in the swelling rate (τ/τ_0_ ≤ 1); the exception is processing by pulses at frequency of 125 Hz with amplitude 500 mV. Note also that the frequency of 100 Hz looks special, since for this frequency the values of τ/τ_0_ for 100 and 500 mV are very close for all investigated samples. In addition, for all investigated liquids, an increase in τ/τ_0_ is observed in the range 100–125 Hz for both pulse amplitudes.

These results show that the treatment of liquid samples according to the specified protocol leads to significant changes in the swelling rate of the membrane, and the effect of such treatment has a sufficiently long relaxation time (on the order of several tens of minutes). The following mechanisms for realization of this effect can be suggested. First, we cannot completely exclude the possibility of appearing ionic impurities from platinum electrodes during processing. It is clear, however, that in this case the change in the swelling rate should monotonically depend on the amount of energy input during electrolysis, i.e., dependences τ/τ_0_ on the pulse repetition rate (these dependences are shown in Figure 8 must behave monotonously. In addition, it is very important for us that for untreated samples of deionized water and isotonic NaCl and Ringer solutions, the decay times τ_0_ coincide to one another with a good accuracy. Finally, for all studied samples, the pH value does not change before and after treatment. Thus, in our opinion, the contamination mechanism can be excluded.

As is known [25], the characteristic times of relaxation processes in water amount to 10^−11^–10^−12^ s. This is, firstly, the lifetime of the hydrogen bond between water molecules; the breaking of the hydrogen bond occurs due to the rotational Brownian diffusion of molecules. Second, this is the decay time of local pressure/density fluctuation. Assuming that the characteristic size of this fluctuation is δ—10–100 nm, we obtain for the decay time of such fluctuation the estimate τ ~ δ/*c*, where *c* is the speed of sound in water, i.e., τ ~ 10^−11^–10^−10^ s. Thus, it would seem that the processes with relaxation times of the order of several tens of minutes cannot arise in water and aqueous solutions of electrolytes.

At the same time, as has been shown in a number of works (see, for example, the review [26] and references therein), there exist long-lived gas nanobubbles with a diameter of 200–300 nm in the bulk of water and aqueous salt solutions under normal conditions. The specified size is usually observed in experiments based on the technique of dynamic light scattering, see Ref. [27] for more detail. In our previous works [28,29,30] it was shown that if a liquid is saturated with a dissolved gas (for example, atmospheric air) and has an ionic component, then ion-stabilized gas nanobubbles spontaneously arise in it. Stabilization is due to the adsorption of anions on the inner (from the side of gas phase) surface of the nanobubbles. Such nanobubbles were called bubstons (abbreviation from *bubble, stabilized by ions*), see [28]. In this case, the surface tension of the bubstons is completely balanced by the negative electrostatic pressure of the adsorbed ions, and the gas pressure inside the bubstons appears to be equal to the atmospheric pressure. Note that the model of stabilization of nanobubbles in a liquid bulk due to the adsorption of ions onto the nanobubble surface has been recently confirmed, see, for example, [31,32,33].

As follows from the studies [28,29,30], the bubstons are stable both mechanically and diffusionally due to the presence of an ionic impurity. It is clear that the ionic component exists in isotonic NaCl and Ringer solutions. As to deionized water, bubstons are stabilized due to the adsorption of HCO_3_^−^ and CO_3_^2−^ anions on the bubston surface, see our recent work [34]. These anions arise owing to the dissociation of carbonic acid, which, in turn, is the result of a hydrolysis of CO_2_ in water. The nucleation of bubstons occurs due to the Coulomb instability of the so-called “droplets of an ionic condensate”. The nuclei for such droplets are the dimers “ion-gas molecule”, see our recent work [35]. In the same work, in particular, we estimated the bubston nucleation time, which is about 2.4 × 10^−8^ s.

When considering the stabilization of bubstons due to ionic adsorption, it is necessary to take into account that negatively charged (due to the adsorption of anions) gas cores of bubstons are always surrounded by a spherically symmetric diffusion cloud of counterions (Debye screening). When bubstons move in a viscous liquid, the peripheral layers of this cloud are effectively “washed away”, which leads to the formation of ξ-potential. In this case, bubstons appear to be negatively charged. Indeed, it is known (for example, see [36,37,38]) that during electrophoresis the bubbles move to the anode.

Let us now estimate the value of the bubston displacement inside a planar capacitor, to which a series of electrical pulses of voltage *U* = 500 mV and duration τ′ = 1 µs with a pulse repetition rate of 10 Hz are applied. Assuming that ξ-potential *φ* of nanobubbles with a radius *R* is expressed as:(5)$$\varphi =\frac{Q}{4\pi \varepsilon \varepsilon_{0}R}$$
where *ε* = 80 is dielectric permittivity of water, and substituting φ = −10 mV (see, for example, [33]), and *R* = 100 nm, we obtain *Q* ~ 10^−17^ C, i.e., about 100 elementary charges. Assuming further the distance *d* between the capacitor plates to be equal to 1 cm, we obtain for the electric field strength inside the capacitor *E* = *U*/(*εd*) ≈ 0.6 V/m. In case of a uniform moving of a bubston, having velocity *v*, we arrive at:(6)QE=6πηRv
where *η* = 10^−3^ Pa⋅s is dynamic viscosity of water at room temperature. Thus, we obtain that force *F* = 0.6 × 10^−17^ N is applied to the bubston, and the bubston velocity is *v* = 0.3 × 10^−8^ m/s, i.e., for a time τ′ = 1 µs, the bubston travels a distance Δ*l*′ ~ 3 × 10^−15^ m. Let us further estimate a random displacement of the bubston Δ*l*″ as a result of Brownian diffusion within the intervals τ′; we obtain:(7)$$\Delta l^{\prime \prime }=\sqrt{D\tau^{\prime }}=\sqrt{\frac{kT\tau^{\prime }}{6\pi R\eta }}\approx 1.4\cdot 10^{-9}\;m$$
where *D* is the Brownian diffusion coefficient, *T* = 300 K. We can see that Δ*l*″ >> Δ*l*′, i.e., within a pulse of 1 µs-width the average position of an individual bubston practically does not change. We cannot imagine a physical mechanism for the processes with relaxation times about tens of minutes as based on the model of individual bubstons.

It is worth to note that in aqueous salt solutions with a sufficiently high concentration of ions, individual bubstons are capable of coagulating to one another with the formation of bubston clusters, ranging in size from 700 nm–2 μm. Experimental studies of bubston clusters in aqueous solutions of NaCl, carried out with dynamic light scattering, are described in [39,40,41]. As was shown in these works, the cluster phase is manifested at ion concentrations > 0.1 M; it is obvious that isotonic NaCl and Ringer solutions meet this condition, while in deionized water the concentration of ions is less. Indeed, in this water the value of pH = 5.7 ± 0.1 before and after processing, i.e., the ion content is 10^−6^ M.

As was found in the experimental work [39], the volume number density of bubston clusters in 0.1 M NaCl solution is about 2 × 10^5^ cm^−3^. The dependence of the volume number density of individual bubstons vs. the ionic concentration in NaCl solution was studied in [29]. It was found in this work that at ion concentration of 10^−6^ M the volume number density of bubstons is 10^6^ cm^−3^, while at a concentration of 10^−1^ M (physiological solutions) we have 3 × 10^7^ cm^−3^. Thus, the volume number densities of bubstons/bubston clusters differ significantly in deionized water and isotonic solutions. Furthermore, the clusters are not revealed in experiments with dynamic light scattering in deionized water. In addition, the clusters are fractal objects; their fractal dimensions and gyration radii in aqueous NaCl solutions were measured in [29,39,40,41]. Finally, the characteristic lifetime of the clusters was measured in [29]. In this experiment, we first measured the distribution of scatterers over size in 1 M NaCl solution, and then the liquid sample was settled for 6 months under stationary conditions in hermetically sealed (without an access of atmospheric air) cell of 2-cm height. It turned out that the micron-sized scatterers (bubston clusters) disappear after settling, while submicron-sized scatterers (individual bubstons) remain in the liquid (“survive”). The disappearance of the coarse scatterers is obviously associated with a cluster floating up following by its destruction at the liquid interface, whereas individual bubstons have a neutral buoyancy, i.e., do not float up. Thus, we can argue that bubstons are an equilibrium phase of an aqueous electrolyte solution under normal conditions, while bubston clusters are a long-lived phase.

Coagulation of bubstons and the cluster formation were theoretically considered in [42]. In this work, the distribution *Q*(*r*) of the entire charge *Q*_0_ of the gas core of radius *R*_0_ and the cloud of counterions surrounding the gas core (the thickness of this cloud is equal to (*r−R*_0_), where the radius vector *r* is originated at the bubston center) in salt solutions with a low ionic concentration is a monotonic function: *Q*(*r*) → 0 for *r →* ∞ for all values of *r*. At the same time, at high ionic concentrations, the function *Q*(*r*) is not monotonic anymore: at a certain value *r = r*_0_, we have *Q*(*r*_0_) = 0 (isoelectric point), but the condition *Q*(*r*) → 0 at *r* → ∞ is still met. Thus, we can talk about the inversion of the sign for the function *Q*(*r*) in concentrated ionic solutions. This effect is described, for example, in Refs [43,44,45]; the physical nature of this effect is related to the gas-core charge *Q*_0_: at low ion concentrations, we have the condition:(8)$$\frac{Q_{0}^{2}}{4\pi \varepsilon_{0}R_{0}}\ll kT,$$
where *k* is the Boltzmann constant. At the same time, at high ionic concentrations, condition (8) is violated:(9)$$\frac{Q_{0}^{2}}{4\pi \varepsilon_{0}R_{0}}\geq kT,$$
which leads to the inversion effect, see [43,44,45]. As was shown in [42], at high ionic concentrations we always deal with two types of compound particles, having electrical charges of the opposite sign. These compound particles are composed of the gas core (the value of its charge is fixed) and a cloud of counterions, remaining after washing off the peripheral ionic layers due to moving in a viscous liquid. This results in Coulomb attraction of oppositely charged compound particles with the formation of clusters.

Based on the results of [42], we modified the hierarchical model of clustering of spherical particles [46]; the hierarchical model was described in detail in the review [47]. In [46] we numerically simulated the growth of clusters composed of spherical monomers. Series of clusters were generated iteratively, starting with *N* individual spherical particles. At each step of the iterative procedure, two clusters were randomly selected, and then these clusters were linked to one another with a random mutual orientation, thus forming a new cluster. We found a solution to the inverse scattering problem in the form of a stochastic ensemble of 2 × 10^3^ hierarchical-type clusters obeying exponential distribution *p*(*N*)–exp(−*αN*) over the number *N* of bubstons in a cluster (*N* ≥ 1, *α* > 0). Here, *α* is a model parameter that takes into account the attraction of particles during the aggregation of a ballistic type. In such a hierarchical model of cluster-cluster aggregation, the average fractal dimension *D* of the ensembles of clusters monotonically depends on the parameter *α*. The parameter *α* can be considered as an additional “degree of freedom”, which allows one to describe the experimentally found angular dependence of the scattering indicatrix elements, see our study [39]. In this work we calculated a set of the scattering matrices as the averages over random cluster ensembles with the statistical parameters, taken on a uniform discrete grid. The solution to the inverse scattering problem was found by minimizing the divergence between the measured angular profiles of the scattering matrix and the same profiles. In Figure 9 we exhibit an example of such realizations for a physiological NaCl solution; *α* = −1.4, fractal dimension of the cluster *D* = 2.45, number of bubstons in the cluster *N* = 420, bubston radius *R* = 100 nm; this figure was taken from [39].

In [46], the electrostatic field strength, at which the effects of coagulation appear, is assumed to be spherically symmetric. The electrostatic force leading to the cluster growth is given by the formula:(10)$$F^{\prime }=\frac{Qq}{4\pi \varepsilon_{0}\varepsilon r^{2}},$$
where *r* is the distance between the center of the bubston with charge *Q* ~ 10^−17^ C and the attractive center on the cluster surface. Let this center have charge *q*; the exact value of this charge is unknown to us. However, without loss of generality, we can put *q* ≈ *−Q*, assuming we deal with an attractive interaction. Thus for *r* ~ 1 µm (the characteristic radius of the cluster), we obtain *F*′ ~ 10^−14^ N >> *F*, where *F* is the force applied to an individual bubston in the field of a flat capacitor, see the comments to Equation (6). It is clear, however, that the estimate for the force *F*′ is very approximate, since we do not know the effective charge *q*. Apparently, a more accurate theoretical analysis and new experiments are required. These experiments should involve the study of the angular dependences of the scattering matrix elements for processed/non-processed liquid samples. It is clear, however, that in the absence of an applied pulsed field, the aggregation of bubstons and the formation of clusters develop in a centrally symmetric field, while upon processing with electric pulses, aggregation of bubstons occurs in a combination of centrally symmetric field and a uniform field of a flat capacitor. Thus it can be argued that the modes of aggregation of bubstons and the fractal properties of bubston clusters will be slightly different for the processed/non-processed samples. In addition, since the volume number density of bubston clusters in deionized water and aqueous salt solutions are different, the effects of treatment with electric pulses for these liquids should also differ from one another. Of course, we still cannot comment on why a change in the geometrical properties of bubston clusters, whose volume number density in 0.1 M NaCl solution is not very high (~2 × 10^5^ cm^−3^), leads to a change in the swelling rate of the polymer membrane. However, we believe that this model could shed light to the mechanisms of a long-term relaxation in water and aqueous salt solutions.

## 5. Conclusions

To summarize, during low-frequency treatment with electric pulses of deionized water and isotonic NaCl and Ringer solutions, the internal parameters of the liquid matrix, which are responsible for the swelling rate of the Nafion polymer membrane, should change. The effect could be associated with a change in the geometric characteristics of bubston clusters during processing. Since bubston clusters are a long-lived phase of water and aqueous salt solutions, within the framework of the proposed model, the long-term relaxation of liquid samples after treatment with electric pulses finds its natural explanation. Since, according to our assumption, Nafion’s membrane is similar to a cell membrane, see [16], the electromagnetic treatment can be used in biomedical practice to prepare drugs with a variable rate of drug penetration into the cell/living tissue.

## Figures and Tables

**Figure 1 polymers-13-01443-f001:**
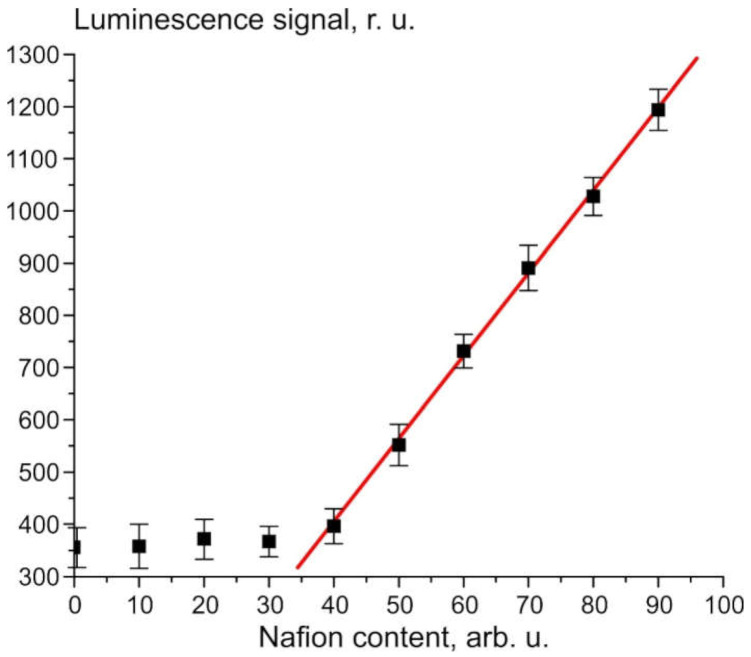
Dependence of the luminescence signal *P* on the content of Nafion in isopropanol solution.

**Figure 2 polymers-13-01443-f002:**
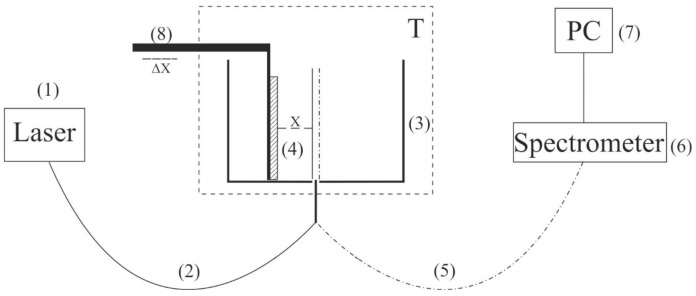
Schematic of the experimental setup for laser luminescence spectroscopy. (1) 369-nm pump beam, (2) optical fiber for pump radiation, (3) thermostat, (4) Nafion plate, (5) optical fiber for photoluminescence signal, (6) spectrometer, (7) personal computer, (8) micrometric-feed table.

**Figure 3 polymers-13-01443-f003:**
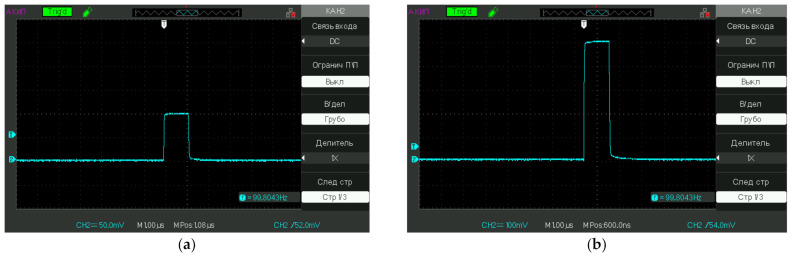
Oscillograms of pulses used in the experiment; the repetition frequency *f* is indicated in the lower right corner (in this particular case *f* = 100 Hz). (**a**)—100 mV and (**b**)—500 mV.

**Figure 4 polymers-13-01443-f004:**
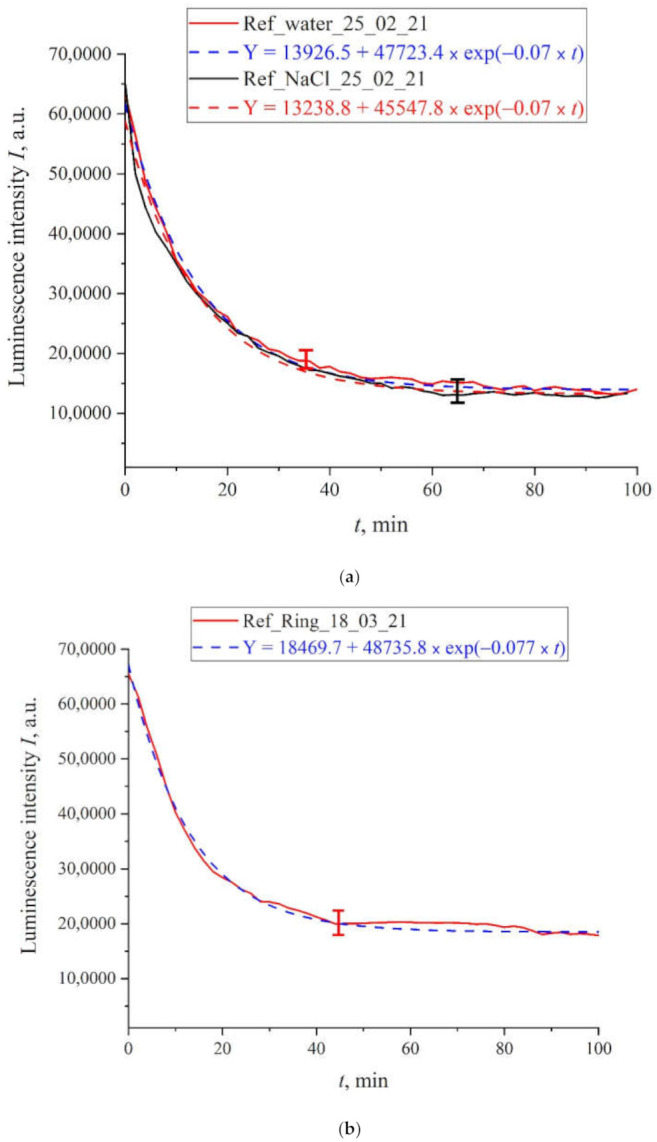
Intensity of luminescence in the spectral maximum vs. the time of soaking of the polymer membrane for untreated solutions (reference samples); (**a**)—deionized water; —NaCl solution; (**b**)—Ringer’s solution.

**Figure 5 polymers-13-01443-f005:**
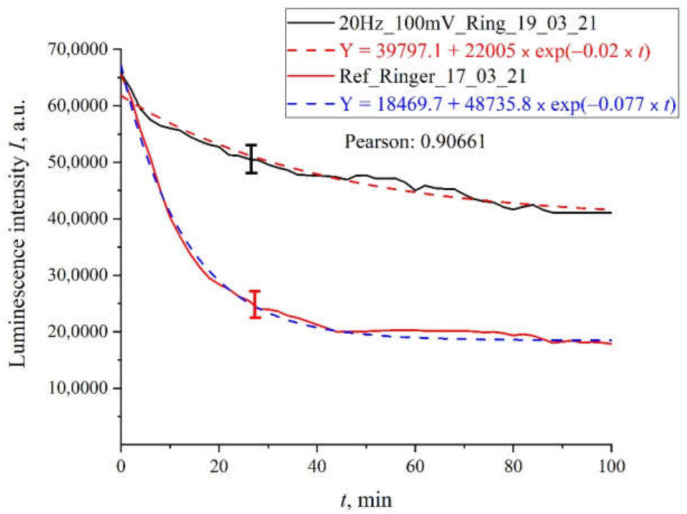
The results of processing with pulses of amplitude 100 mV at a repetition rate of 20 Hz: for Ringer’s solution.

**Figure 6 polymers-13-01443-f006:**
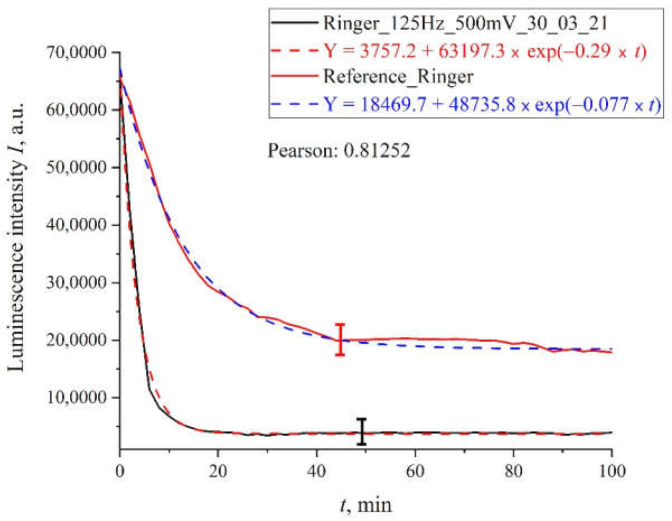
The results of processing with pulses of amplitude 125 mV at a repetition rate of 500 Hz: for Ringer’s solution.

**Figure 7 polymers-13-01443-f007:**
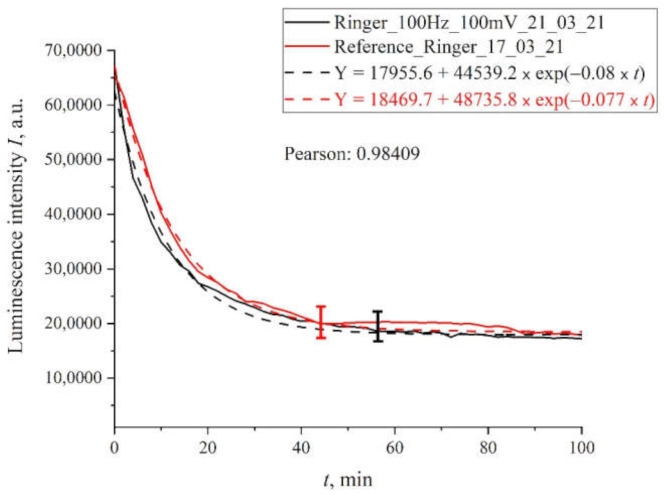
The results of processing with pulses of amplitude 100 mV at a repetition rate of 100 Hz for Ringer’s solution.

**Figure 8 polymers-13-01443-f008:**
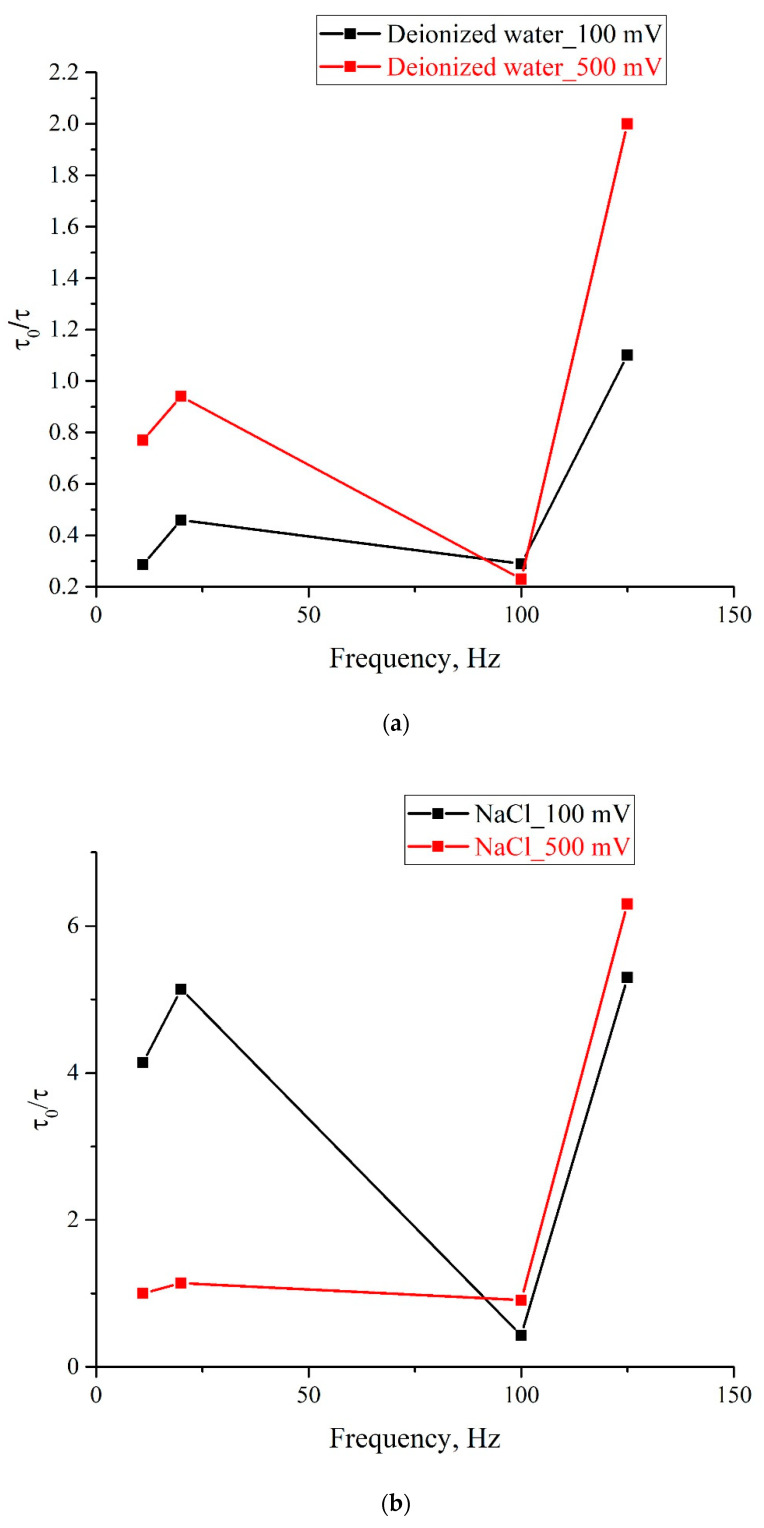
Dependence of τ/τ_0_ on the pulse repetition rate for pulse amplitudes of 100 mV (black curve) and 500 mV (red curve); (**a**)—deionized water; (**b**)—NaCl solution; (**c**)—Ringer’ solution.

**Figure 9 polymers-13-01443-f009:**
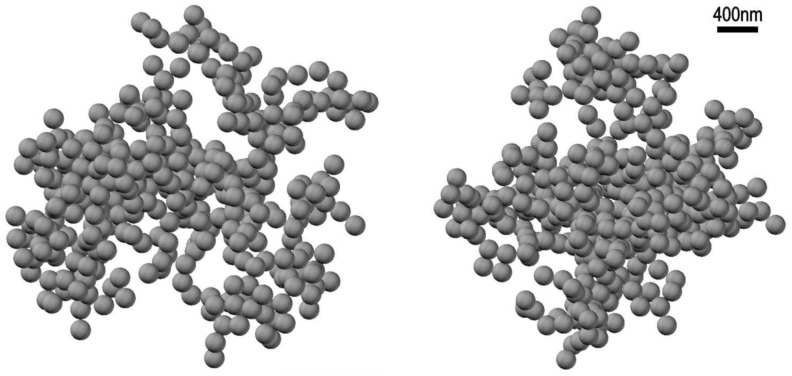
Mutually perpendicular projections of a stochastic realization of hierarchic bubston cluster with the parameters: the number of bubstons *N* = 400, the bubston radius *R* = 100 nm, fractal dimension *D* = 2.45, see [39].

## Data Availability

The data presented in this study are available on request from the corresponding author.

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
