# Peer review of "Long-Term Effect of Low-Frequency Electromagnetic Irradiation in Water and Isotonic Aqueous Solutions as Studied by Photoluminescence from Polymer Membrane"

_polymers, 2021, doi:10.3390/polym13091443_

Round 1

Reviewer 1 Report

The study of interactions of nonionizing electromagnetic waves with whole organisms, and specifically with well-characterized cellular model systems in vitro, has received increased recognition: A growing number of experimental findings has been reported, and hypothetic mechanisms which might be involved in mediating these effects have emerged. It is worth to mention here that stress genes are also induced by low frequency magnetic fields. The cellular response to magnetic fields is activated by unusually weak stimuli, and involves pathways only partially associated with heat shock stress. Since magnetic fields interact with moving charges, as we have shown in enzymes, it is possible that magnetic fields stimulate the stress response by interacting directly with moving electrons in DNA. In this article, the swelling of a polymer membrane Nafion in deionized water and isotonic NaCl and Ringer’s solutions was studied by photoluminescent spectroscopy. Although the topic in this work was interesting, the presentation in this manuscript was very poor. This manuscript should be rejected for published in Polymers. However, if the authors are willing to make the substantial revisions according to my comments, I would be glad to re-review this manuscript. Here are my detailed comments:

  1. The detailed literature review indicates efforts made by the authors. The coherence of the related work, however, is still not clear. It may help the authors by answering the following questions: Why are these works relevant? Which specific problems were addressed? How are the previous results related with the latest work? What are the outstanding, unresolved, research issues? Which of them has been solved by the proposed study? Answering the questions leads to the novelty of the proposed work naturally. Besides, the current one is nothing but a literature review. Why their work is important comparing to previous reports? I think this is essential to keep the interest of the reader.
  2. Materials and Methods part, Although the results look “making sense”, the current form reads like a simple lab report. The authors should dig deeper in the results by presenting some in-depth discussion.
  3. In Figs. 4, 6, 9 and 11, the authors should give the explanations for the difference of data collected from different sources.
  4. We used a series of pulses with a repetition rate of 11 - 125 Hz; the pulse amplitudes were equal to 100 and 500 mV. It turned out that at certain pulse repetition rates and their amplitudes, the characteristic swelling time of the polymer membrane significantly differs from the swelling time in untreated (reference) samples. At the same time, there is no effect for certain frequencies / pulse amplitudes. The time interval between electromagnetic treatment and measurements was about 20 minutes. Thus, that in our experiments the effects associated with the long-term relaxation of liquids on the electromagnetic processing are manifested. The authors should give some explanation on above conclusions and data.
  1. Polymer membrane has been widely used in the industry. Polymer membrane has been applied in a number of practical applications, for example fibrous porous media, (see [A fractal model for capillary flow through a single tortuous capillary with roughened surfaces in fibrous porous media, Fractals, 2021, 29(1):2150017; Fractals, 2019, 27(7): 1950116]). Authors should introduce some related knowledge to readers. I think this is essential to keep the interest of the reader.
  2. Although the results look “making sense”, the authors should dig deeper in the results by presenting some in-depth discussion, such as implications of the results, such as possible application of them.

Author Response

We are grateful to the referee for carefully reading the manuscript and making comments. We agree with these comments. Indeed, the previous version of the manuscript looked more like a report on the work of our laboratory, rather than a scientific message. This annoying oversight has been corrected. We have completely rewritten the Introduction section, removing unnecessary literary references from there (we left only monographs as the main literary sources). In addition, we significantly reduced the number of figures and completely removed all tables, leaving only meaningful results. We believe that the main result of our work is the discovery of the effects of long-term relaxation in water and aqueous solutions of electrolytes after treatment with electrical impulses with a low repetition rate. In principle, this type of processing is used in medical practice, see, for example, reference [9] in the new version of the manuscript. In the updated version, we have presented a qualitative theoretical model that explains the effect of long-term relaxation; this model is based on the presence of nanobubble clusters in water and aqueous electrolyte solutions. The numerical estimates presented in the new version of the manuscript show that after processing with electric pulses, effects with a sufficiently long relaxation time can indeed arise.

Reviewer 2 Report

The manuscript "Long-term effect of low-frequency electromagnetic irradiation in water and isotonic aqueous solutions as studied by photoluminescence from polymer membrane" is not well presented and there some serious flaws. In general the introduction well written and some expect if reading a well organized manuscript. The author basically shown in experimental part 2 the results so then the real result part is showing lots of figures of same measurements but there no text given for such. If showing scientific results especially for electromagnetic irradiation some scientific explanation should be discussed or given. The authors have to rewrite and represent the result part with not listing each case seperatly which confuse really. Table 1-8 should be shown in one and maximal 2 Tables and mean values with standard deviation should be listed. Figure 4-11 are shown in general in supplementary if no text given. Please make 1-3 Figures out of it at maximum.

The result part said results and discussion while discussion part is shown as well.

Past research addressed such topic in a more scientific way.

David E. Moilanen, Ivan R. Piletic, and Michael D. Fayer, Water Dynamics in Nafion Fuel Cell Membranes: The Effects of Confinement and
Structural Changes on the Hydrogen Bond Network, J. Phys. Chem. C 2007, 111, 8884-8891

While such topic surely interesting the authors have to present their results in a more clear form avaoiding many figures and tables. So far there need be some references and discussion either under results or seperated presented.

The scientific depth is missing as well the application where such new knowledge can adapt to

There also many typos in the script with words wrote together as well dots missing etc. Please correct such.

Author Response

We are grateful to the referee for carefully reading the manuscript and making comments. We have significantly reduced the text by removing unnecessary figures and tables. We referred to an article that was recommended by a reviewer. We've also eliminated annoying typos. Finally, we rewrote the text of the manuscript, inserting into the new version a qualitative theoretical model explaining the effects of long-term relaxation in water and aqueous salt solutions, subjected to treatment with electrical impulses with a low repetition rate.

Round 2

Reviewer 1 Report

It is ok.

Reviewer 2 Report

The authors gave good answers as well the manuscript improved after revision. Therefore accept as it is.